# Unconsciousness and amnesia after cross-body electric shocks not involving the head–A prospective cohort study

**Karin Biering⬚\*, Anette Kærgaard, Ole Carstensen, Kent J. Nielsen⬚**

Department of Occupational Medicine–University Research Clinic, Danish Ramazzini Centre, Goedstrup Hospital, Herning, Denmark

\* karbie@rm.dk

## Abstract

### Introduction

Little is known about how electrical current passes through the human body except that it follows the physical rule of least resistance. Whether organs remote from the shortest route of the current can be affected is unknown, as different types of tissue vary in resistance. This may explain why some people exposed to electrical injury report symptoms from the central nervous system (CNS). In this study, we examined the association between exposure to cross-body electrical current and immediate CNS symptoms.

### Material and methods

In a prospective cohort study, we followed 6960 members of the Danish Union of Electricians for 26 weeks using weekly questionnaires. We identified 2356 electrical shocks, and for each shock we asked whether the exposure was cross-body or same-side. We excluded those who reported exposure to the head as well as those who could not report the entry and exit points of the current. We examined two outcomes: becoming unconscious or having amnesia of the event. We use percentages to describe the data and logistic regression to analyze the results.

### Results

We found that unconsciousness and amnesia following electric shocks were rare events (0.6% and 2.2%, respectively). We found an increased risk of reporting unconsciousness and amnesia in those exposed to cross-body electrical shock exposure compared to those with same-side exposure (Odds Ratio 2.60[0.62 to 10.96] and Odds Ratio 2.18[0.87 to 5.48]).

### Conclusion

Although the outcomes investigated are rare, we cannot rule out a possible effect on the CNS when persons are exposed to cross-body electrical current even though it does not pass through the head.

**Data Availability Statement:** Data cannot be shared publicly because of the very few persons with the outcomes, that makes anonymisation impossible. Data are available from Redcap Aarhus

University Institutional Data Access (contact via arbejdsmedicin@goedstrup.rm.dk) for researchers who meet the criteria for access to confidential data.

**Funding:** KB, AK, OC and KJN received a shared grant from the Danish Working Environment Research Fund, grant number 22-2017-09 Website: https://amff.dk/about-the-fund/ The funders had no role in study design, data collection and analysis, decision to publish, or preparation of the manuscript.

**Competing interests:** The authors have declared that no competing interests exist.

## Introduction

Exposure to electrical current is known to affect the central nervous system (CNS) in terms of mental and cognitive problems following electrical injuries. Symptoms such as headache, migraine, concentration problems, memory problems and fatigue are frequently reported following exposure to electrical current [1–3]. Whether these symptoms are related to the physiological impact of the electrical current or are a reaction to the traumatic event is unknown [4]. Previous studies have often been retrospective, imposing a risk of selection bias and recall bias, and are based on populations recruited from burn units, which means that the injuries were typically rather severe.

Morse et al. describe the theoretical current pathway as the linear path from entry point to exit point, but they also refer to studies that report symptoms remote from this theoretical pathway [5]. The explanation for this could be that the current does not follow just the shortest linear line, but also follows other paths depending on the resistance in the tissue and the surface. Previous studies of current passing through the body in different paths have focused on ventricular fibrillation [6].

We examined the association between exposure to cross-body electrical current at entry points and exit points remote from the head and immediate CNS symptoms in an observational prospective study.

We hypothesize that if the current crosses the spinal cord, it may also affect the cerebrum and thus cause CNS symptoms.

## Material and methods

### Material

A prospective cohort study following 6960 members of the Danish Union of Electricians for 26 weeks from October 2019 using a weekly web-based questionnaire. Initially, a baseline questionnaire was answered covering health, working conditions and demographics. In the following 26 weeks, the electricians received a short questionnaire that covered health symptoms and asked them to report whether they had experienced electrical shock during the past week. If they had experienced a shock, the questionnaire provide space to include details regarding the shock (voltage, entry and exit points, severity, dry/wet conditions, physical injury, contact to health services, unconsciousness and amnesia). The data collection and a general description of the shocks are presented in Biering et al. 2021 [7].

### Methods

We examined the association between cross-body exposure to the electrical current and the immediate impact on the CNS in terms of reporting loss of consciousness and amnesia among the electricians who had experienced shocks.

For each reported shock, cross-body exposure was identified through the question: "Did the current leave the body on the same or the opposite side of where it entered?" Answer categories were "same side" (non-exposed), "opposite side" (exposed) and "do not know". We excluded those who answered "do not know- or had not answered and those where entry or exit point of the shock was reported as either "head" or "do not know" or was missing.

The outcome was defined based on the participants' reporting of being unconscious or having amnesia of the event, under the assumption that this reflected the immediate impact on the CNS. We defined three different outcomes based on the following questions: "Did you lose consciousness in connection with the shock?" and "Did you experience amnesia in connection with the shock?" Answer categories were "yes", "no" or "do not know" for both questions. The

outcomes were unconsciousness, amnesia and a combination of unconsciousness and/or amnesia.

We used percentages to describe the data according to whether the shock resulted in the outcome or not. Furthermore, we used logistic regression to examine the association between the exposure and the outcomes. Results were presented as odds ratios with 95% confidence intervals. We conducted two additional analyses: one where we adjusted for self-reported severity, as we were in doubt whether severity could be a confounder (it could be a consequence if experiencing the outcome), and one where we included those who answered "do not know" to the questions regarding unconsciousness and amnesia because they may have experienced the outcomes but were not able to recall the event.

Each person could experience several shocks during follow-up, and we considered these events to be independent observations, although related to the same person.

Data was analyzed using Stata 17.0.

### Ethics

The study was approved by the Danish Data protection Agency via Central Region Denmark #1-16-02-139-19. All participants gave written informed consent, prior to participation.

The Danish rules for which research projects that must be reported to a scientific ethics committee are clearly stated in The Committees Act. All health science studies are subject to notification, which means that every health science research project must be reported to the scientific ethics committee system, cf. §14 of the Committees Act. However, there are exceptions, cf. the Committees Act §14 subsection 2–5., which means that research projects, referred to in these exceptions, do not have to be reported to a scientific ethics committee. Questionnaire surveys are referred to in §14 subsection 2, where it states that questionnaire studies only have to be reported to the scientific ethics committee system, if the project includes human biological material. This implies that this study does not need to be reported to the committee.

### Results

In total, 2356 electrical shocks were reported by 1612 electricians. Exclusion of electricians who were unable to provide information on the entry and exit points of the current left 1235 in the sample, 233 in the exposed group (cross-body shock) and 1002 in the non-exposed group (same-side shock). The most common entry point was hands/fingers (96%/$n$ = 1180), and accordingly the most common exit point was also hands/fingers (79%/$n$ = 973).

Both unconsciousness and amnesia were rare events among the electricians experiencing shocks (0.6%/$n$ = 8 and 2.2%/$n$ = 21). We found that all shocks that led to unconsciousness or amnesia were experienced by men (but very few female electricians participated in the study), and there was no clear pattern related to age. Same-side exposure was more common than cross-body exposure. Those who reported unconsciousness or amnesia often also reported that the shock was somewhat severe or worse (Table 1). Due to confidentiality issues, we do not report subgroups with less than five persons, and as only 27 persons reported unconsciousness/amnesia in connection with the shock, we could not report whether there were more shocks with high voltage, burn wounds, contact to emergency rooms and with no-let-go.

We found an increased risk of reporting affected in those with cross-body exposure compared to those without CNS symptoms (OR: 2.20 [0.97 to 4.95] (Table 2). In this group there was a slightly higher estimate for unconsciousness (OR:2.60 [0.62 to 10.96]) than amnesia (OR:2.18 [0.87 to 5.48]). The estimates were slightly reduced when we adjusted for self-reported severity and if we included those who did not know/recall whether they experienced

**Table 1. Demographics and cross-body exposure to electrical current in relation to unconsciousness and amnesia in connection with electrical shock reported by Danish electricians (*n* = 1235).**

|  | Not affected CNS | Affected CNS | Subgroups of affected CNS | |
|---|---|---|---|---|
|  | *n* = 1208 | *n* = 27 | | |
|  |  |  | Unconsciousness | Amnesia |
| Male *n* (%) | 1190 (98.5%) | 27 (100.0%) | 8 (100.0%) | 21(100.0%) |
| Female *n* (%) | 18 (1.5%) | 0 (0.0%) | 0 (0.0%) | 0 (0.0%) |
| Age >40 years *n* (%) | 469 (38.8%) | 10 (37.0%) | 2 (25.0%) | 8 (38.1%) |
| Age ≤40 years *n* (%) | 739 (61.2%) | 17 (63.0%) | 6 (75.0%) | 13 (61.9%) |
| Somewhat severe or worse *n* (%) | 51 (4.2%) | 5 (18.5%) | 3 (37.5%) | 4 (19.0%) |
| Not severe or a little severe *n* (%) | 1157 (95.8%) | 22 (81.5%) | 5 (62.6%) | 17 (81.0%) |
| Cross-body exposure *n* (%) | 224 (18.5%) | 9 (33.3%) | 3 (37.5%) | 7 (33.3%) |
| Same-side exposure *n* (%) | 984 (81.5%) | 18 (66.7%) | 5 (62.5%) | 14 (66.7%) |

*Affected CNS is a combination of reporting of unconsciousness and/or amnesia. Two persons reported both outcomes

unconsciousness or amnesia in the outcomes (Table 2). The risk estimates were not statistically significant.

## Discussion

In a prospective cohort study of 6960 Danish electricians, we identified 2356 electrical shocks during 6 months' follow-up. Among the 1235 who had shocks that met the criteria for analysis in this study (sufficient data on the entry and exit points of the current), 1208 did not report symptoms from the CNS, whereas 27 did (eight with unconsciousness and 21 with amnesia, two reported both). We found that cross-body exposure to electrical current increased the risk of reporting symptoms from the CNS (OR: 2.20 [0.97 to 4.95]). Risk of reporting unconsciousness (OR: 2.60 [0.62 to 10.96]) was higher than that of reporting amnesia (OR: 2.18 [0.87 to 5.48]). This may indicate that electrical current passing through the human body between two contact points remote from the head and across the body can affect the CNS. To our knowledge, this is the first study to examine this association. A few limitations should be mentioned. First, the limited number of persons with unconsciousness and amnesia, despite the large cohort followed for 6 months and the large number of reported shocks. This limitation may have had negative effect on the precision of the estimates. In previous studies, reporting of symptoms from the CNS was frequent, but previous literature was based on an injured population, and did not report all types of electrical shocks as we did in our study. Many of the participants (36%) were unable to report the exit point of the current, and thus, it was not possible to determine whether cross-body exposure had taken place. In an additional analysis, we adjusted

**Table 2. Sensitivity analysis of the association between cross-body exposure and affected CNS in terms of unconsciousness and/or amnesia.**

|  | Affected CNS* | Unconsciousness | Amnesia |
|---|---|---|---|
|  | OR [95%CI] | OR [95%CI] | OR [95%CI] |
| Same-side exposure (reference) | 1 | 1 | 1 |
| Cross-body exposure | 2.20 [0.97 to 4.95]. | 2.60 [0.62 to 10.96] | 2.18 [0.87 to 5.48] |
| Cross-body exposure adjusted for severity | 1.67 [0.71 to 3.95]. | 1.38 [0.30 to 6.28] | 1.66 [0.63 to 4.40] |
| Cross-body exposure with inclusion of those uncertain of the outcome | 1.68 [0.80 to 3.54]. | 1.92 [0.59 to 6.31] | 1.94 [0.83 to 4.53] |

OR: Odds Ratio 95%CI: 95% confidence interval

Affected CNS*: Unconsciousness and/or amnesia

for severity, but whether or not this could be considered a confounder is dubious because the outcomes (unconsciousness and amnesia) would be considered as high in severity and thus part of the outcome. Although the study was prospective, both the exposure (path of the current) and the outcomes (unconsciousness and amnesia) were reported cross-sectionally, but the respondents were not aware of the hypotheses and did not always describe the shocks in sufficient detail. Another possible explanation for the outcomes could be that the electrical shock was due to an epileptic seizure. However, none of those who reported either unconsciousness or amnesia suffered from epilepsy. Recall bias is most likely not a problem for the outcomes, but could have affected the reporting of exposure because amnesia and/or unconsciousness could affect the ability to report details regarding the shock. But if this were the case, the participant would most likely end up in the non-exposed group because of the absent information regarding entry and exit points, and this would have had an effect on the sample size of this study and possibly underestimate the association. We made additional analyses where we included those who were uncertain whether they had experienced CNS symptoms, and this reduced the estimates slightly.

If the impact on the CNS in terms of unconsciousness and amnesia were not caused by the electrical shock itself but by the trauma of receiving a shock, then we would not find an association between cross-body exposure and the outcomes. We were not able to identify similar studies, and thus we were not able to compare our findings with previous findings, but several other studies and reviews report that both neurological and mental symptoms can occur following exposure to electrical shock [1–3, 8, 9]. However, these reports build almost solely on case studies and other retrospective designs, whereas the prospective studies lack detailed information regarding the shock. The present study may contribute to the debate regarding whether electrical current can affect the CNS even if the direct entry or exit point is not through the head. Although the analysis is based on a small sample, we conclude that cross-body electrical exposure remote from the head can affect the CNS.

## Acknowledgments

We wish to thank the members of the Danish Union of Electricians for their participation in the study.

## Author Contributions

**Conceptualization:** Karin Biering, Anette Kærgaard, Kent J. Nielsen.

**Data curation:** Karin Biering, Kent J. Nielsen.

**Formal analysis:** Karin Biering.

**Funding acquisition:** Karin Biering, Anette Kærgaard, Ole Carstensen, Kent J. Nielsen.

**Investigation:** Karin Biering, Anette Kærgaard.

**Methodology:** Karin Biering, Anette Kærgaard.

**Project administration:** Kent J. Nielsen.

**Writing – original draft:** Karin Biering.

**Writing – review & editing:** Anette Kærgaard, Ole Carstensen, Kent J. Nielsen.

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
