## [Decision Letter · Decision Letter 0]

31 Oct 2022

PONE-D-22-22946Unconsciousness and amnesia after cross-body electric shocks not involving the headPLOS ONE

Dear Dr. Biering,

Thank you for submitting your manuscript to PLOS ONE. After careful consideration, we feel that it has merit but does not fully meet PLOS ONE’s publication criteria as it currently stands. Therefore, we invite you to submit a revised version of the manuscript that addresses the points raised during the review process.

Your manuscript has been assessed by two peer-reviewers and their reports are appended below.  The reviewers comment that the article requires more detail on the study design and the methodology. In addition, the reviewers comment that the discussion section should be strengthened by a careful comparison of the presented results with the published literature, and discuss how the current findings could affect future studies and policy making.  Could you please carefully revise the manuscript to address all comments raised?

We look forward to receiving your revised manuscript.

Kind regards,

Maria Elisabeth Johanna Zalm, Ph.D

Editorial Office

PLOS ONE

Journal Requirements:

2. In ethics statement in the manuscript and in the online submission form, please provide additional information about the patient records/samples used in your retrospective study. Specifically, please ensure that you have discussed whether all data/samples were fully anonymized before you accessed them and/or whether the IRB or ethics committee waived the requirement for informed consent. If patients provided informed written consent to have data/samples from their medical records used in research, please include this information.

Reviewers' comments:

Reviewer's Responses to Questions

**Comments to the Author**

1. Is the manuscript technically sound, and do the data support the conclusions?

Reviewer #1: No

Reviewer #2: Yes

2. Has the statistical analysis been performed appropriately and rigorously? 

Reviewer #1: No

Reviewer #2: Yes

3. Have the authors made all data underlying the findings in their manuscript fully available?

Reviewer #1: Yes

Reviewer #2: No

4. Is the manuscript presented in an intelligible fashion and written in standard English?

Reviewer #1: No

Reviewer #2: No

5. Review Comments to the Author

Reviewer #1: The authors address in a satisfactory way the gap of the examined issue at the introduction section but they should clearly refer the aim of their study at the last paragraph as well as the hypothesis in different sentence .

The methodology design (experimental tests, measurements etc) needs more details. The methodology has important details missing. Subsection for materials and methodology could help the flow.

The results of the study are presented in a satisfactory way

The authors should present the main findings of their study at the first paragraph of the discussion section and also compare their findings to similar studies of the literature. Discussion section is to poor. The authors discuss their findings with no other studies.

Reviewer #2: The manuscript addresses the topic whether immediate CNS symptoms (i.e. unconsciousness, amnesia) might occur in electrical shocks without direct entry or exit point on the head. I appreciate the hypothesis of the study that electric currents flowing through the body and crossing the spinal cord may induce CNS symptoms, even if the head (brain) is not directly affected. This is the first study on this topic. The main strength of the study is the study population because it is based on a prospective cohort of electricians with weekly follow-up.

The data collection and a general description of the shocks are presented in a previous study of the authors including details on the type of electrical shock in terms of current types (AC versus DC) and voltages (low versus high voltage). I recommend to present additionally the numbers for the current types and voltages and to discuss the proportions (e.g., in Table 1) in the present manuscript. Thus, it may be possible to associate the circumstances of the electrical shocks and the outcome further.

Additionally, there are some minor points:

Title: ‘a prospective cohort study’ should be added as an important feature of the study.

The abstract should be structured with the headers ’Introduction’, ‘Material and Methods’, ‘Results’ and ‘Conclusion’.

Line 40: Please provide the risk estimate (i.e. odds ratio (OR).

Line 56: A dot is missing at the end of the sentence.

Line 63: Please add the year in which the study was conducted.

Line 70: Please move the comma to ‘same side’ (non-exposed), ‘opposite side’.

Lines 71-74: These two sentences should be moved to the result section.

Line 83: The risk estimate and confidence interval (95%?) should be provided.

Line 93 and Line 95: Please add the numbers to facilitate reading.

Line 94: A dot is missing at the end of the sentence.

Lines 99: Please delete the dot before the parenthesis.

Lines 101-102 Header of the Table 1: ‘Danish electricians (n=1235)’ is a bit misleading, because 1,235 is the number of events, not of the electricians.

Table1: There is probably a mistake with the numbers in the table because the four columns of gender, age, severity and exposure do not sum up to 1,235, but to 1,264. Please check it. - In column ‘unconsciousness’ and row ‘somewhat severe or worse’ and ‘Not severe or a little severe’ is an error because the sum is not 100%.

Lines 103 and 112: Is the explanation of ‘Combined’ beneath the table correct or should the logical operator only be ‘and’ instead of ‘and/or’?

Line 104-107: Please provide the risk estimates and note the non-significant confidence intervals.

Line 105: Please delete the dot before the parenthesis.

Line 109 Table 2: Please add [CI95%] after OR in the second row and add the abbreviation below the table.

Line 114: Please start the discussion with a sentence on the key results (i.e., number of persons and events).

Line 155: Please use ‘Ann N Y Acad Sci.’ to be consistent.

The manuscript lacks from several punctuation and may benefit from some closer proofreading by a native speaker.

6. PLOS authors have the option to publish the peer review history of their article (what does this mean?). If published, this will include your full peer review and any attached files.

Reviewer #1: No

Reviewer #2: No

---

## [Author Response · Author response to Decision Letter 0]

7 Feb 2023

Dear Editor and reviewers,

Thank you very much for valuable corrections, comments and suggestions.

We have now revised the manuscript according to the reviews and specific answers to each point are presented below. We hope you find the provided changes and answers satisfactory. 

On behalf of the authors

Karin Biering

Review Comments to the Author

Reviewer #1: The authors address in a satisfactory way the gap of the examined issue at the introduction section but they should clearly refer the aim of their study at the last paragraph as well as the hypothesis in different sentence.

Authors reply: This is now rewritten to highlight the aim and hypothesis clearly

The methodology design (experimental tests, measurements etc) needs more details. The methodology has important details missing. Subsection for materials and methodology could help the flow.

Authors reply: We have added these subheadings and provided more detail regarding the study, that we originally left out to avoid self-plagiarism 

The results of the study are presented in a satisfactory way

The authors should present the main findings of their study at the first paragraph of the discussion section and also compare their findings to similar studies of the literature. Discussion section is to poor. The authors discuss their findings with no other studies.

Authors reply: We have added the main findings to the beginning of the discussion, and have tried to elaborate the discussion. However, comparing our findings to previous studies is difficult as mentioned by reviewer #2, this is the first study on this topic, so direct comparison is not possible..

Reviewer #2: The manuscript addresses the topic whether immediate CNS symptoms (i.e. unconsciousness, amnesia) might occur in electrical shocks without direct entry or exit point on the head. I appreciate the hypothesis of the study that electric currents flowing through the body and crossing the spinal cord may induce CNS symptoms, even if the head (brain) is not directly affected. This is the first study on this topic. The main strength of the study is the study population because it is based on a prospective cohort of electricians with weekly follow-up.

The data collection and a general description of the shocks are presented in a previous study of the authors including details on the type of electrical shock in terms of current types (AC versus DC) and voltages (low versus high voltage). I recommend to present additionally the numbers for the current types and voltages and to discuss the proportions (e.g., in Table 1) in the present manuscript. Thus, it may be possible to associate the circumstances of the electrical shocks and the outcome further.

Authors reply: Thank you for these kind words. We have elaborated on the methods section as also reviewer#1 suggested to provide more information regarding the study. We would also have provided more details regarding the circumstances of the shock. However, in when using data from Statistics Denmark it is custom not to report subgroups containing less than 5 persons, so not all details could be reported in the group with affected CNS in table 1. Instead, we have now added a sentence including these characteristics without presenting the exact numbers in the results section

Additionally, there are some minor points:

Title: ‘a prospective cohort study’ should be added as an important feature of the study.

Authors reply: This is added to the Title

The abstract should be structured with the headers ’Introduction’, ‘Material and Methods’, ‘Results’ and ‘Conclusion’.

Authors reply: The abstract has been provided with the suggested headers

Line 40: Please provide the risk estimate (i.e. odds ratio (OR).

Line 56: A dot is missing at the end of the sentence.

Line 63: Please add the year in which the study was conducted.

Line 70: Please move the comma to ‘same side’ (non-exposed), ‘opposite side’.

Lines 71-74: These two sentences should be moved to the result section.

Line 83: The risk estimate and confidence interval (95%?) should be provided.

Line 93 and Line 95: Please add the numbers to facilitate reading.

Line 94: A dot is missing at the end of the sentence.

Lines 99: Please delete the dot before the parenthesis.

Lines 101-102 Header of the Table 1: ‘Danish electricians (n=1235)’ is a bit misleading, because 1,235 is the number of events, not of the electricians.

Authors reply: All of these suggestions has been added/changed

Table1: There is probably a mistake with the numbers in the table because the four columns of gender, age, severity and exposure do not sum up to 1,235, but to 1,264. Please check it. - In column ‘unconsciousness’ and row ‘somewhat severe or worse’ and ‘Not severe or a little severe’ is an error because the sum is not 100%.

Authors reply: The numbers are actually correct, but we realize that the presentation of them were not elegant. In the original Table 1, we present three columns of outcome (unconsciousness(n=8), amnesia(n=21) and a combination of unconsciousness and/or amnesia (n=27, two persons reported both outcomes). This means that the last column include all with either unconsciousness or amnesia and those with both. We have made a new layout of the table and added a longer footnote, to make this clear. 

Lines 103 and 112: Is the explanation of ‘Combined’ beneath the table correct or should the logical operator only be ‘and’ instead of ‘and/or’?

Authors reply: Please refer to our reply regarding table 1 above

Line 104-107: Please provide the risk estimates and note the non-significant confidence intervals.

Line 105: Please delete the dot before the parenthesis.

Line 109 Table 2: Please add [CI95%] after OR in the second row and add the abbreviation below the table.

Line 114: Please start the discussion with a sentence on the key results (i.e., number of persons and events).

Line 155: Please use ‘Ann N Y Acad Sci.’ to be consistent.

Authors reply: All of these suggestions has been added/changed

The manuscript lacks from several punctuation and may benefit from some closer proofreading by a native speaker.

Authors reply: We have now had professional British proof-reading, thus several minor changes in wording has been changed on top of the suggestions provided by the reviewers.

---

## [Decision Letter · Decision Letter 1]

21 Mar 2023

Unconsciousness and amnesia after cross-body electric shocks not involving the head – a prospective cohort study

PONE-D-22-22946R1

Dear Dr. Biering,

We’re pleased to inform you that your manuscript has been judged scientifically suitable for publication and will be formally accepted for publication once it meets all outstanding technical requirements.

Kind regards,

James Mockridge

Staff Editor

PLOS ONE

Additional Editor Comments (optional):

Reviewers' comments:

Reviewer's Responses to Questions

**Comments to the Author**

1. If the authors have adequately addressed your comments raised in a previous round of review and you feel that this manuscript is now acceptable for publication, you may indicate that here to bypass the “Comments to the Author” section, enter your conflict of interest statement in the “Confidential to Editor” section, and submit your "Accept" recommendation.

Reviewer #1: All comments have been addressed

Reviewer #2: All comments have been addressed

2. Is the manuscript technically sound, and do the data support the conclusions?

Reviewer #1: Yes

Reviewer #2: Yes

3. Has the statistical analysis been performed appropriately and rigorously? 

Reviewer #1: Yes

Reviewer #2: Yes

4. Have the authors made all data underlying the findings in their manuscript fully available?

Reviewer #1: Yes

Reviewer #2: Yes

5. Is the manuscript presented in an intelligible fashion and written in standard English?

Reviewer #1: Yes

Reviewer #2: Yes

6. Review Comments to the Author

Reviewer #1: No more comments. Thank you for your nice work and the Submission to PlosOne. All issues have been addressed in a satisfactory way.

Reviewer #2: (No Response)

7. PLOS authors have the option to publish the peer review history of their article (what does this mean?). If published, this will include your full peer review and any attached files.

Reviewer #1: No

Reviewer #2: No

---

## [Editor Report · Acceptance letter]

24 Mar 2023

PONE-D-22-22946R1 

Unconsciousness and amnesia after cross-body electric shocks not involving the head – a prospective cohort study 

Dear Dr. Biering:

I'm pleased to inform you that your manuscript has been deemed suitable for publication in PLOS ONE. Congratulations! Your manuscript is now with our production department. 

Kind regards, 

on behalf of

Dr James Mockridge 

Staff Editor

PLOS ONE